# Designing of pH-Sensitive Hydrogels for Colon Targeted Drug Delivery; Characterization and In Vitro Evaluation

**DOI:** 10.3390/gels8030155

**Published:** 2022-03-03

**Authors:** Muhammad Suhail, Yu-Fang Shao, Quoc Lam Vu, Pao-Chu Wu

**Affiliations:** 1School of Pharmacy, Kaohsiung Medical University, Kaohsiung City 80708, Taiwan; u108830004@kmu.edu.tw; 2Department of Biomedical Science and Environmental Biology, Kaohsiung Medical University, Kaohsiung City 80708, Taiwan; s26104030@gs.ncku.edu.tw; 3Department of Clinical Pharmacy, Thai Nguyen University of Medicine and Pharmacy, 284 Luong Ngoc Quyen Str., Thai Nguyen 24000, Vietnam; vuquoclam@tump.edu.vn; 4Department of Medical Research, Kaohsiung Medical University Hospital, Kaohsiung 80708, Taiwan; 5Drug Development and Value Creation Research Center, Kaohsiung Medical University, Kaohsiung 80708, Taiwan

**Keywords:** hydrogels, chitosan, β-cyclodextrin, drug release, 5-aminosalicylic acid

## Abstract

In the current research work, pH-sensitive hydrogels were prepared via a free radical polymerization technique for the targeted delivery of 5-aminosalicylic acid to the colon. Various proportions of chitosan, β-Cyclodextrin, and acrylic acid were cross-linked by ethylene glycol dimethacrylate. Ammonium persulfate was employed as an initiator. The development of a new polymeric network and the successful encapsulation of the drug were confirmed by Fourier transform infrared spectroscopy. Thermogravimetric analysis indicated high thermal stability of the hydrogel compared to pure chitosan and β-Cyclodextrin. A rough and hard surface was revealed by scanning electron microscopy. Similarly, the crystallinity of the chitosan, β-Cyclodextrin, and fabricated hydrogel was evaluated using powder X-ray diffraction. The swelling and drug release studies were performed in both acidic and basic medium (pH 1.2 and 7.4, respectively) at 37 °C. High swelling and drug release was observed at pH 7.4 as compared to pH 1.2. The increased incorporation of chitosan, β-Cyclodextrin, and acrylic acid led to an increase in porosity, swelling, loading, drug release, and gel fraction of the hydrogel, whereas a decrease in sol fraction was observed. Thus, we can conclude from the results that a developed pH-sensitive network of hydrogel could be employed as a promising carrier for targeted drug delivery systems.

## 1. Introduction

The most common types of chronic inflammatory bowel diseases (IBD) are ulcerative colitis (UC) and Crohn’s disease (CD), both having the same signs and symptoms. Both UC and CD lead to other digestive disorders and inflammation in the digestive system. The main cause of these two disorders is still unknown, but a number of factors are involved in the prevalence of CD and UC. Some of these are the geographical location, an unbalanced diet, genetics, and an improper response of the immune system. The diagnosis of both diseases is found to be higher in urban areas, while in rural areas both diseases are diagnosed rarely in comparison. Both diseases have their adverse effects and challenges; however, patients still enjoy a good quality of life. It is a fact that these diseases occur more often at a younger age, and thus disturbs the patient’s half-life. IBD is going to be a major health problem in future, even in developing countries [1]. Hence, different strategies and conventional therapies, including sulfasalazine, 5-aminosalicylic acid (5-ASA), corticosteroids, thiopurines, and methotrexate, are advocated for both CD and UC management. 5-ASA is considered the drug of choice among other conventional therapies for the treatment of IBD due to its safety and high efficacy [2].

5-ASA, an anti-inflammatory drug, is commonly used for the treatment of CD and UC. The benefit of 5-ASA usage is not only to manage an IBD patient’s suffering, but also to protect them from the development of colon cancer [3]. The elimination half-life of 5-ASA ranges between 30–90 min [4]. The absorption of 5-ASA occurs very rapidly from the small intestine, whereas a very small quantity is present in the colon. Thus, due to the short half-life and rapid absorption, repeated doses of 5-ASA are required during the day to maintain the therapeutic level for a long period. The frequent intake of 5-ASA leads to the generation of some severe side effects such as pancreatitis, hepatitis, blood dyscrasias, pleuropericarditis, and interstitial nephritis [5]. Hence, various drug carrier systems have been developed in order to overwhelm the side effects linked to multiple intakes of 5-ASA. Ahmad et al. (2021) prepared 5-ASA-loaded gelatin nanoparticles (NPs) coated with eudragit-S100 for localized release of 5-ASA [6]. Stavarache and his coworkers prepared 5-ASA loaded chitosan–carrageenan hydrogel beads for the treatment of IBD [7]. Likewise, Bautzová et al. (2012) developed chitosan pellets by an extrusion and spheronization method for the controlled release of 5-ASA [8]. However, a lot of work is still required in order to overwhelm the side effects produced as a result of multiple administrations of 5-ASA. Due to the diverse features and prominent characteristics of hydrogels, the authors tried to develop chitosan (CS) and beta-cyclodextrin (β-CD)-based pH-sensitive hydrogels for the targeted delivery of 5-ASA to the colon in a controlled way.

Hydrogels are three-dimensional cross-linked polymeric networks holding the ability to absorb a greater amount of water without losing their original geometry. An excellent swelling index, stability, drug loading, biocompatibility, degradability, gel ability, and gelling capabilities are the special properties of hydrogels which enhance their use in the pharmaceutical and biomedical fields [9,10]. Recently, due to an excellent response to external stimuli such as pH, ionic strength, and temperature, a special type of hydrogel, known as a stimuli-sensitive hydrogel, has gained great interest, especially for the release of a drug in response to a stimulus. A pH-sensitive hydrogel is a special type of stimuli-sensitive hydrogel employed for the controlled and targeted delivery of a drug to a specific site within the gastrointestinal tract [11].

CS is a cationic, biodegradable, biocompatible, and non-toxic biopolymer. It consists of a b-(1-4)-2-acetamido-2-deoxy-D-glucose unit, which is the main alkaline deacetylation product of chitin [12]. This natural, polysaccharide polymer contains hydroxyl and amine groups, which are known as good cross-linkable sites for polymerization with other various polymers or monomers in order to develop drug carrier systems with improved thermal stability, mechanical strength, desirable swelling, and a favorable release profile for drugs in a stable polymeric network of hydrogels [13]. β-CD is an acyclic oligosaccharide, which has an inner hydrophobic cavity with an outer hydrophilic rim. The US Food and Drug Administration (FDA) have listed both β-CD and γ-CD as safe chemicals that can be used for different purposes. β-CD has played an important role on an industrial and pharmaceutical level as it is easily available and economical and has different dimensions of cavity, which are suitable for a number of drug candidates [14].

Here, we reported on the preparation, characterization, and evaluation of newly prepared pH-sensitive hydrogels of CS and β-CD cross-linked with acrylic acid (Aa). This system was used for the targeted delivery of 5-ASA to the colon. The novelty of the current research work was based on the incorporation of CS and β-CD with Aa, which led to the development of pH-sensitive hydrogels. The prepared hydrogels were subjected to a series of characterizations and studies including Fourier transform infrared spectroscopy, thermogravimetric analysis, scanning electron microscopy, powder X-ray diffraction, sol-gel analysis, dynamic swelling, and percent drug release. Both swelling and drug release studies were performed in a low acidic pH of 1.2 and a high basic pH of 7.4 in order to determine the pH sensitive nature of the fabricated hydrogels. The sensitivity of the developed hydrogel was mostly based on the incorporation of acrylic acid with both CS and β-CD. The pKa value of a reagent plays an important role in its protonation and deprotonation. The reagent will protonate at a pH below its pKa value and deprotonate at a pH above its pKa value. The pKa value of Aa is near to 4, hence the protonation and deprotonation of the acrylic acid occurred at pH 1.2 and 7.4, respectively. Consequently, the swelling and drug release was found to be a maximum of pH 7.4 as compared to pH 1.2, demonstrating the pH-sensitive nature of the prepared hydrogel. A very small amount of 5-ASA reaches the colon due to its high absorption in the upper gastrointestinal tract; hence, an insufficient amount of the drug reaches the colon. Thus, multiple doses of 5-ASA are needed to be taken in a day, which leads to severe side effects. Therefore, the authors prepared a pH-sensitive hydrogel that has the potential to target a high amount of 5-ASA into the colon via a controlled fashion and overcome the side effects generated as a result of multiple dose intakes.

## 2. Results and Discussion

### 2.1. Preparation of CS/β-CDcPAa Hydrogels

A free radical polymerization technique was employed for the synthesis of the CS/β-CDcPAa hydrogels. Nine formulations with different compositions of CS, β-CD, and Aa were cross-linked (Table 1) and novel pH-sensitive hydrogels were prepared for the targeted delivery of 5-ASA. The formulated hydrogels indicated high gelation with the increased incorporation of CS, β-CD, and Aa. Hence, the prepared hydrogels were subjected to a series of studies for further investigation. The proposed schematic diagram and the physical appearance of the developed hydrogels are given in Figure 1A,B. The CS/β-CDcPAa hydrogels were made of CS, β-CD, and Aa; therefore, their shape or swelling was affected by the pH of the medium. Low swelling was observed at pH 1.2 due to the protonation of the functional groups of polymers and monomer, while maximum swelling was observed at pH 7.4 due to the deprotonation. Thus, a pH-responsive nature was exhibited by the fabricated hydrogel, which is the most suitable for the targeted delivery of a drug such as 5-ASA.

### 2.2. FTIR Analysis

A Fourier transform infrared spectroscopy (FTIR) analysis is performed in order to understand the structural configuration of the reagents and prepared drug carrier system [15]. Hence, spectroscopy was conducted for the confirmation of the CS/β-CDcPAa hydrogel preparation. Figure 2 indicates the FTIR spectrum of CS, β-CD, and Aa; the unloaded CS/β-CDcPAa hydrogel; 5-ASA; and the loaded CS/β-CDcPAa hydrogel. The FTIR spectra of CS (Figure 2A) revealed symmetrical stretching vibrations of the amine NH group by peaks at 3442 and 2860 cm^−1^ connected with a pyranose ring (OH and CH_2_). The stretching vibration of carbonyl (amide-I), N–H (amide-II), and C–N (amide-III) was detected at 1652, 1598, and 1380 cm^−1^, respectively [16]. Similarly, the stretching vibration of ether C=O, methyl and methylene C−H, and alcoholic O−H was observed at 1016, 2965, and 3310 cm^−1^, respectively, by the FTIR spectra of β-CD (Figure 2B) [14]. Peaks at 1596, 1670, and 2892 cm^−1^ were assigned to the stretching vibration of C–C, C–O, and –CH_2_, respectively, while the stretching vibration of the –C–O group was observed at 1205 by the FTIR spectrum of Aa (Figure 2C) [17,18]. Due to the chemical interaction among the hydrogel contents, i.e., CS, β-CD, and Aa, a fluctuation was seen in the position of their certain peaks, similarly to the peaks of CS and β-CD, which were changed from 1598, 1380 cm^−1^ and 1016, 2965 cm^−1^ to 1615, 1410 and 1028, 2998 cm^−1^ peaks of fabricated hydrogel, respectively (Figure 2D). Similarly, a few peaks of Aa were modified from 1205, 1596, to 1242, 1613 cm^−1^ peaks of prepared hydrogels. Certain peaks of polymers and monomer disappeared while a few new ones were formed as indicated by FTIR spectral analysis of unloaded CS/β-CDcPAa hydrogel. This shifting, disappearance, and formation of new peaks indicated the successful cross-linking of Aa over the backbone of both CS and β-CD, thus we can conclude from the FTIR spectrum of the developed hydrogel that a new polymeric network of hydrogel was developed. Similarly, FTIR spectrum of 5-ASA (Figure 2E) exhibited characteristic peaks at 1648 and 1630 cm^−1^ indicated stretching and bending of COOH and NH_2_. Likewise, two peaks at 1358 and 1456 indicated stretching and bending of C–N and –OH group, respectively [19]. A few peaks of 5-ASA were altered slightly from 1456 and 1630 cm^−1^ to 1470 and 1605 cm^−1^ of drug loaded CS/β-CDcPAa hydrogel as indicated in Figure 2F. The change in position of 5-ASA peaks was due to its encapsulation by developed hydrogel without any interaction with hydrogel contents [20].

### 2.3. Sol–Gel Analysis

Sol–gel analysis was performed to determine the soluble and insoluble fraction of developed hydrogels. The soluble or uncross-linked fraction of hydrogel is known as a sol fraction, whereas the insoluble or cross-linked fraction of hydrogel is known as gel fraction [21]. Both sol and gel fractions were influenced by the various concentrations of incorporated hydrogel contents, i.e., CS, β-CD, and Aa as shown in Table 2. Gel fraction was increased with the increasing concentration of CS and β-CD. The reason may be the availability of a greater number of free radicals for Aa content. During the polymerization process, free radicals were generated by CS and β-CD, thus as the concentration of both CS and β-CD was increased, an increase in generation of free radicals was observed. This led to fast and rapid polymerization of polymers content with monomer, and thus an increase in gel fraction was observed. Similarly, an increase in free radicals and polymerization process was observed with the increasing concentration of Aa; as a result, cross-linking among hydrogel contents was increased, and thus an increase in gel fraction was detected and vice versa. Khanum et al. (2018) prepared polymeric hydrogels and reported an increase in gel fraction with the increasing composition of polymer and monomer [22], which further supports our investigation. Unlike gel fraction, a decrease in sol fraction was observed because both sol and gel fraction are inversely proportional to one another. An increase in gel fraction leads to a decrease in sol fraction and vice versa [23].

### 2.4. Porosity Study

Porosity study is performed in order to evaluate the penetration of a medium into the drug carrier system [24]. An important role is played by porosity especially in swelling and drug loading of hydrogels. Channels are generated, through which water penetration of hydrogel networks occurs. The porosity of a hydrogel is influenced by various concentrations of its contents, i.e., CS, β-CD, and Aa as indicated in Figure 3A–D. Porosity was increased as the concentration of CS, β-CD, and Aa was increased. The reason may be attributed to the viscosity of the reaction mixture. High concentration of polymers and monomer led to a viscous reaction mixture, due to which the evaporation of bubbles was stopped. Thus, as a result, interconnected channels were generated. These channels increased the porosity of hydrogel and vice versa. Hence, we can conclude that high swelling and drug loading is dependent on porosity of hydrogel [25]. *p*-values were found to be less than 0.05 for all formulations of the fabricated hydrogel.

### 2.5. Thermogravimetric Analysis (TGA)

TGA is conducted to determine the thermal stability of the reagent and fabricated system individually [26]. Weight loss of individual reactant, i.e., CS and β-CD and formulated hydrogel at different temperature range was determined by their TGA thermogram as indicated in Figure 4. TGA thermogram of CS (Figure 4A) indicated a weight loss of 10% at 120 °C due to the evaporation of water. Onward temperature led to further loss of weight as almost 40% loss was detected at temperature 310 °C. After that, degradation of pure CS was begun and continued until complete degradation. Similarly, TGA thermogram of β-CD (Figure 4B) exhibited initial weight loss of 12% at 115 °C due to the loss of bounded water. A further increase in temperature up to 380 °C led to weight loss of 63%. Finally, degradation of β-CD was started at 385 °C [27]. Likewise, a TGA thermogram (Figure 4C) of developed hydrogel depicted a very minute weight loss of 2–3% at 200 °C correlated with water loss of polymers. After that, a weight loss of 75% was observed as temperature increased to 485 °C. After that, degradation of formulated hydrogel begun and continued until entire degradation. Thus, it can be concluded from a TGA thermogram of individual reactants and formulated hydrogel that thermal stability of pure CS and β-CD was enhanced due to cross-linking and grafting. The formulated hydrogel indicated a greater thermal stability as compared to unreacted CS and β-CD. Greater thermal stability of fabricated hydrogel showed strong inter-molecular interaction of contents which existed as a result of cross-linking, grafting, and polymerization [28,29,30]. Barkat and his co-worked prepared chondroitin sulfate based hydrogel and reported increase in thermal stability of the fabricated hydrogel compared to its reagent [31], which further supports our hypothesis.

### 2.6. Powder X-ray Diffraction (PXRD) Analysis

XRD analysis is conducted to understand the form change in comparison to its initial form [32]. Therefore, PXRD analysis was conducted in order to investigate the physical state of CS, β-CD, and formulated hydrogel. PXRD diffractogram of CS, β-CD, and formulated hydrogel is shown in Figure 5. CS exhibited intense, sharp, and prominent crystalline peak at 2θ = 19.52° and 38.90°, whereas PXRD analysis of β-CD indicated its crystalline structure at 2θ = 12.40°, 16.20°, 22.86°, and 28.64°, as indicated in Figure 5A,B. The high intense, sharp and crystalline peaks of both CS and β-CD were disappeared in PXRD analysis of fabricated hydrogel (Figure 5C). The high intensity and sharp crystalline peaks were replaced with decreased and low intensity peaks, which indicated the successful grafting of polymers with monomer, due to which the original crystalline structures of CS and β-CD were changed [33]. Lee et al. (2007) developed amphiphilic poly (l-lactide)-grafted chondroitin sulfate copolymer and reported decrease in crystallinity of the hydrogel contents [34].

### 2.7. Scanning Electron Microscopy (SEM)

The main purpose of conducting SEM was to evaluate the surface morphology of the fabricated CS/β-CDcPAa hydrogel as indicated in Figure 6. A hard and irregular surface with minor pores was exhibited by the developed hydrogel. Large wrinkles and cracks can be seen in Figure 6, which may be attributed to partial collapsing of the gel during the dehydration. The hard surface of the fabricated hydrogel indicated the strong intermolecular interaction existed among the hydrogel contents after the polymerization process. The penetration of water occurs through the pores, which causes the swelling of the hydrogel. The pore size almost remained the same at low pH values due to the protonation of the hydrogel contents, i.e., CS, β-CD, and Aa, while at high pH values an increase in pore size was observed due to the generation of repulsive forces among the same functional groups of the hydrogel contents. Hence, a greater amount of water penetrated the hydrogel network and thus maximum swelling was achieved [35].

### 2.8. Dynamic Swelling Studies

Swelling indicates the water absorption capability of a developed hydrogel system [36]. Hence, swelling studies were carried out for developed hydrogel in order to evaluate its pH-sensitive nature in two different pH values, pH 1.2 and 7.4. Greater swelling was achieved at pH 7.4 as compared to pH 1.2 as shown in Figure 7A,E. The low swelling in pH 1.2 was due to the protonation of NH_2_ group of CS. However, due to the grafting of CS with other polymeric networks, a significant decrease in the number of NH_2_ groups of CS was perceived [37]. Similarly, COOH groups of Aa remained un-ionized at acidic pH 1.2 and form conjugate with counter ions, and thus strong hydrogen bonding was formed between carboxylate groups, which strengthened the hydrogel network thus preventing the water penetration into polymeric hydrogel. As a result the hydrogel remained collapsed and insignificant swelling was perceived at pH 1.2 and vice versa. The maximum swelling of fabricated hydrogel at pH 7.4 was due to the ionization or deprotonation of COOH groups of Aa. The pKa value of Aa is almost 4, hence the protonation and deprotonation of a reagent depends on its pKa value. The reagent will be protonated at pH below its pKa value and will be deprotonated at pH above its pKa value. Due to the deprotonation of COOH groups of Aa, charge density was increased, which led to the generation of strong electrostatic repulsive forces. These forces repelled each other and thus greater swelling was achieved at pH 7.4 [38].

As well as pH, swelling of the hydrogel was also influenced by various concentrations of its contents. An increase in swelling was perceived at both pH 1.2 and 7.4 with the increasing concentration of CS and β-CD, indicated in Figure 7B,C. An increase in CS concentrations led to the generation of a greater amount of NH_2_ groups. As a result, charge density was increased, thus strong repulsive forces were produced. These forces caused in higher swelling of hydrogels. Thus, we can say that increase in CS concentration led to an increase in hydrophilicity of hydrogel and vice versa [39]. Similarly, an increase in β-CD concentration led to an increase in the swelling of the hydrogel. The reason may be the availability of a high number of functional groups, i.e., –OH and CH_2_OH. Thus, an increase in thr generation of –OH and CH_2_OH was depicted with the increasing concentration of β-CD, due to which charge density was increased and thus an increase in swelling was perceived [40]. Like CS and β-CD, an increase in swelling was perceived with the increasing concentration of Aa (Figure 7D) and vice versa [41].

### 2.9. Polymer Volume Fraction

Polymer volume fraction was estimated for the fabricated hydrogel at both pH 1.2 and 7.4, as indicated in Table 1. Polymer volume fraction was found low at pH 7.4 while greater at pH 1.2. The varying concentrations of hydrogel contents, i.e., CS, β-CD, and Aa, have greatly affected the polymer volume fraction. A drop was seen in polymer volume values with the increased incorporation of CS, β-CD, and Aa, which may be interlinked with the high swelling index of the developed hydrogel. The high and low polymer volume values at pH 1.2 and 7.4 demonstrated the excellent swelling capabilities of the developed hydrogel at high pH value of 7.4 [42].

### 2.10. Drug Loading Analysis

Drug loading indicates the amount of drug encapsulated by polymeric network of hydrogel [43]. The quantification of drug loaded by CS/β-CDcPAa hydrogel was determined by weight and extraction methods as shown in Table 2. Swelling plays a vital role in the loading of drugs. An increase or decrease in swelling leads to a respective increase or decrease in the loading of drugs because there is a direct relation between swelling and drug loading [44] and vice versa. Drug loading was affected highly by the different concentrations of CS, β-CD, and Aa. Drug loading was increased with the increasing concentration of polymers and monomer. An increase in CS, β-CD, and Aa concentration led to an increase in generation of their functional groups, due to which charge density increased, and as a result swelling increased. Thus, an increase in drug loading was observed and vice versa.

### 2.11. Dissolution Studies

The rate of drug release from the fabricated hydrogel at a regular intervals of time is determined by dissolution studies [45]. Like swelling, the pH-responsive nature of CS/β-CDcPAa hydrogel was investigated in both pH 1.2 and 7.4. Drug release at pH 1.2 was found to be very low compared to pH 7.4, as indicated in Figure 8A. The reason may be the protonation and deprotonation of carboxylic group of Aa. At low pH 1.2, COOH groups of Aa form conjugate with counter ions and strengthen the polymeric network of the hydrogel. Thus, penetration of water was restricted, and low swelling and drug release was observed. Similarly, the NH_2_ group of CS remained protonated, which also led to low swelling and drug release at pH 1.2. On other hand, as the pH enhanced from 1.2 to 7.4, deprotonation of COOH groups of Aa occurred, which led to high charge density. Thus, high swelling and drug release was detected at pH 7.4 due to the presence of strong repulsive forces, which swelled the polymeric hydrogel networks [46,47]. Drug release studies were also carried out for commercial available product Pentasa (500 mg) as indicated in Figure 8B. A drug release of more than 98% was observed at pH 7.4 within an initial 4 h, while at pH 1.2, a drug release of 95% was detected within an initial 8 h. Hence, comparing the drug release of commercial product and the fabricated hydrogel, we can demonstrate that fabricated hydrogel has targeted the delivery of 5-ASA to colon effectively in a controlled fashion. Thus, the novel pH-responsive developed CS/β-CDcPAa hydrogel could be used as a suitable agent for targeted drug delivery systems.

Drug release was influenced by the varying concentrations of CS, β-CD, and Aa as shown in Figure 8C–E. An increase in swelling and drug release was seen with the increasing concentration of hydrogel contents. The possible reason may be the availability of a greater number of functional groups of the hydrogel contents, which led to high charge density, thus higher swelling and drug release was observed [48,49,50].

### 2.12. Kinetic Modeling

Swelling index is the most important feature of formulated CS/β-CDcPAa hydrogel, which influences the loading and release of drugs. The regression co-efficient is represented by “r” value. “r” values of all kinetic models are given in Table 3, and the most suitable model was selected among the all kinetic models on the closeness of “r” value to 1. Thus, we can easily predict from Table 3 that “r” values of first order were nearby to 1 as compared to “r” values of other kinetic models. Therefore, we can demonstrate that first order of kinetic models was followed by all formulations of the fabricated hydrogel. “n” value determines the type of diffusion, i.e., Fickian diffusion if 0.45 ≤ n, whereas non-Fickian or anomalous transport analogous to coupled diffusion/polymer relaxation if 0.45 ≤ n ≤ 0.89. “n” values were confirmed within the range of 0.5228–0.6310 for all formulation of the developed CS/β-CDcPAa hydrogel, thus demonstrating a non-Fickian diffusion [51,52].

## 3. Conclusions

CS/β-CDcPAa hydrogels were synthesized successfully by the cross-linking of various compositions of CS, β-CD, and Aa through a free radical polymerization technique. FTIR confirmed the cross-linking of polymers and monomer, and loading of drugs by the fabricated hydrogel. High thermal stability was indicated by TGA, while PXRD showed a reduction in highly intense peaks of both CS and β-CD by the developed hydrogel. A rough and uneven surface was revealed by SEM. A significant swelling and drug release was observed in a basic medium as compared to an acidic medium, indicating the pH-sensitive nature of the fabricated hydrogel. An increase in swelling and drug release was observed with the increasing compositions of polymers and monomer. Similarly, an increase in porosity, drug loading, and gel fraction was detected with the increase in CS, β-CD, and Aa compositions, while sol fraction was decreased. Greater polymer volume fraction at low pH 1.2, while lesser at high pH 7.4, indicated a significant swelling of fabricated hydrogels. The results of our studies suggested that the current prepared pH-sensitive network of hydrogels can be employed as an efficient carrier for targeted delivery of different hydrophilic drugs.

## 4. Materials and Methods

### 4.1. Materials

5-aminosalicylic acid was purchased from Alfa Aesar (Lancashire, UK). Chitosan (MW = 100,000–300,000) and β-Cyclodextrin (MW = 1396 Da) was acquired from Acros organics (Fair Lawn, NJ, USA) and Alfa Aesar (Thermo Fisher Scientific, Ward Hill, MA, USA), respectively. Acrylic acid (MW = 72.06 g/mol) and ammonium persulfate (MW = 228.21) were sourced from Acros (Carlsbad, CA, USA) and Showa (Tokyo, Japan), respectively, whereas ethylene glycol dimethacrylate (MW = 198.22 g/mol) was purchased from Alfa-Aesar (Tewksbury, MA, USA).

### 4.2. Preparation of CS/β-CDcPAa Hydrogels

The cross-linking of various combinations of CS, β-CD, and acrylic acid (Aa) was carried out by the free radical polymerization technique in the presence of ethylene glycol dimethacrylate (EGDMA) and ammonium persulfate (APS), and as a result chitosan/β-cyclodextrin-co-poly(acrylic acid) (CS/β-CDcPAa) hydrogels were prepared. A set of nine formulations is given in Table 1, where various combinations of polymers such as β-CD, chitosan, and monomer Aa were shown with constant compositions of EGDMA and APS. Hence, for the preparation of CS/β-CDcPAa hydrogels, a weighed amount of CS and β-CD was taken separately. CS is not completely soluble in water; hence a 1% acetic acid solution was used for dissolving CS. β-CD was dissolved in deionized distilled water. APS is completely soluble in water, hence APS was dissolved in a specific amount of deionized distilled water and added into the β-CD solution. After a proper mixing, the mixture was added into the CS solution and stirred for 25 min with 50 rpm. After that, Aa was added drop wise into the above mixture. In addition, the mixture of polymers, initiator, and monomer was stirred for 10 min. Finally, cross-linker EGDMA was added into the mixture dropwise. The mixture was stirred until a transparent solution was formed. Nitrogen gas was purged through the solution to remove dissolved oxygen and then the transparent solution was transferred into the glass molds, which were placed in water bath at 55 °C for 2 h. After 2 h, temperature was changed to 65 °C for the next 24 h. After the preparation, discs of 6 mm were formed by cutting the gel. An ethanol and water mixture was used for washing and removing any impurity attached to the surface of the gel discs. The discs were placed in vacuum oven at 40 °C for entire dehydration after exposing them for 24 h at room temperature. The dried discs were processed for further investigation.

### 4.3. Fourier Transform Infrared Spectroscopy (FTIR)

The investigation of drug interaction with hydrogel contents was performed by ATR (Attenuated total reflectance). Hence, spectral analysis of CS, β-CD, Aa, the unloaded CS/β-CDcPAa hydrogel, 5-ASA, and the drug loaded CS/β-CDcPAa hydrogel was performed by ATR. The FTIR spectrum analysis was performed within 4500–500 cm^−1^ range by using NICOLET 380 FTIR (Thermo Fisher Scientific, Ishioka, Japan) [15].

### 4.4. Sol–Gel Analysis

The main purpose of sol–gel analysis was to evaluate the amount of uncross-linked content of fabricated hydrogel by using Soxhlet extraction process. Therefore, weighed hydrogel discs were added into the Soxhlet apparatus containing deionized distilled boiling water. The process of extraction was continued for 10 h. After that, the extracted discs were removed and placed in a vacuum oven at 40 °C until a constant weight was obtained [16]. The following equations were used for the estimation of sol and gel fraction:(1)$$\mathrm{Sol}\;\mathrm{fraction}\;\%=\frac{{\;C}_{1}-{\;C}_{2}}{{\;C}_{2}}\times 100$$
(2)Gel fraction=100−Sol fraction 

C_1_ indicates the initial weight of dried hydrogel disc before the extraction process, and C_2_ is the final weight after the extraction.

### 4.5. Porosity Study

The porosity of CS/β-CDcPAa hydrogel was determined by the solvent displacement method. Absolute ethanol was used as a displacement solvent. Hence, dried weighed discs of the hydrogel (T_1_) were placed for 5 days in absolute ethanol so that maximum ethanol penetrated through the pores into the hydrogel network. After that, discs were removed and weighted again after extra ethanol attached to the surface was wiped away (T_2_) [17]. The given equation was used for the determination of (%) porosity;
(3)$$(\%)\;\mathrm{Porosity}=\frac{T_{2}-T_{1}}{\rho_{V}}\times 100$$

ρ is the density of absolute ethanol and V is the swelling volume of hydrogel discs.

### 4.6. Thermogravimetric Analysis (TGA)

TGA (PerkinElmer Simultaneous Thermal Analyzer STA 8000, (PerkinElmer Ltd., Buckinghamshire, UK)) was conducted for CS, β-CD, and CS/β-CDcPAa hydrogel in order to investigate their thermal stability. Thus, 4–6 mg of the finely grounded sample was taken and placed in an aluminum pan attached to a microbalance. TGA was performed at a rate of 20 °C/min within temperature range of 40–600 °C with a constant flow of nitrogen gas throughout the experiment [18].

### 4.7. Powder X-ray Diffraction (PXRD) Analysis

The physical nature of CS, β-CD, and CS/β-CDcPAa hydrogel was investigated by PXRD. The samples were evaluated by using XRD-6000 Shimadzu, Tokyo, Japan. PXRD spectrum was analyzed within angle range of 10–60° with an enhancing rate of 2°/min [19].

### 4.8. Scanning Electron Microscopy (SEM)

The surface morphology of the cross-linked CS/β-CDcPAa hydrogel was examined by the SEM (JSM-5300 model, (JEOL, Tokyo, Japan). The sample was initially ground, and then with the help of adhesive tape the sample was mounted on an aluminum stub. Gold sputter was employed for the coating of gold beneath argon atmospheric conditions. Finally, micrographs were achieved at different magnification [20].

### 4.9. Dynamic Swelling Studies

pH-sensitivity of CS/β-CDcPAa hydrogel was investigated by dynamic swelling studies in two different pH values of 1.2 and 7.4 at body temperature. Therefore, a dried, weighed disc of hydrogel was immersed in 100 mL of respective buffer solution. After a specified period of time, hydrogel disc was removed, blotted with filter paper, and then weighed again. This process was continued until an equilibrium swelling was achieved where no further increase in the weight of hydrogel disc was observed [21]. This study was performed in a triplicate. The given equation was used for calculation of swelling index:(4)$$\left(q\right)=\frac{{\;S}_{2}}{S_{1}}$$
q represents the dynamic swelling, S_1_ indicates the initial weight of dried hydrogels disc before swelling, and S_2_ reveals the final weight after swelling at time t.

### 4.10. Polymer Volume Fraction

Polymer volume fraction is the polymer fraction in complete swelled state. It is denoted by V_2,s_. Equilibrium volume swelling (Veq) data of fabricated hydrogels at two different pH values (1.2, and 7.4) were used for the estimation of polymer volume fraction [22]. Thus, the given equation was employed for the estimation of polymer volume fraction:V_2,s_ = 1/Veq(5)

Veq indicates the equilibrium volume swelling data employed for the determination of polymer volume fraction.

### 4.11. Drug Loading Analysis

Loading of 5-ASA by formulated hydrogel was performed by a swelling and diffusion method [23]. Briefly, 1% solution of 5-ASA was formed in a phosphate-buffered solution of pH 7.4. Weighed dried discs of the hydrogel were placed in the drug solution for 5 days. After equilibrium swelling and loading, the discs of hydrogels were taken out from the solution of 5-ASA, blotted with filter paper, and washed by distilled water to remove the attached drug with the surface of the hydrogel discs. After that, the discs were subjected to drying in a vacuum oven at 40 °C until a constant weight was achieved.

Two methods were carried out for the determination of loaded drugs by the developed hydrogels. In the extraction method, a solvent replacement procedure was carried out for the elimination of all drugs from the developed hydrogel. Hence, 25 mL buffer solution of pH 7.4 was used within weighed loaded hydrogel discs placed for a specific interval of time. After that, the sample was collected and the medium was replaced by fresh medium of the same quantity. This process was continued until no drug remained inside the hydrogel disc. All the collected samples were then analyzed on UV–Vis spectrophotometer (U-5100,3J2-0014, Tokyo, Japan) at ʎmax 218 nm to determine the loaded contents.

Another method was weight method. The given equation was used for the determination of encapsulated drug by the fabricated hydrogels.
Drug loaded quantity = DL − DUL(6)

DL shows the weight of drug loaded hydrogel discs and DUL represents the weight of unloaded hydrogel discs [24].

### 4.12. Dissolution Studies

Dissolution apparatus II (USP dissolution (Sr8plus Dissolution Test Station, Hanson Research, Chatsworth, CA, USA)) and UV–V is spectrophotometer (U-5100, 3J2-0014, Tokyo, Japan) [25] was employed for evaluation of in vitro drug release from commercial available product Pentasa (500 mg, Ferring Pharmaceutical limited, Vanlose, Denmark) and fabricated CS/β-CDcPAa hydrogel at both acidic and basic medium (pH 1.2 and 7.4), respectively. The Pentasa and loaded hydrogel discs were immersed separately in a 900 mL dissolution medium of both pH 1.2 and 7.4 at 37 ± 0.5 °C and 50 rpm. A 5 mL sample was taken at a specified period of time and fresh medium of the same quantity was added back to maintain the sink condition. The UV analysis of all collected samples was performed at ʎmax 218 nm in a triplicate.

### 4.13. Kinetic Modeling

Different kinetic models such as zero-order, first-order, Higuchi, and Korsmeyer–Peppas models were used for understanding the mechanism and order of drugs from CS/β-CDcPAa hydrogel by fitting the in vitro drug release data into the respective models [26].
Zero order kinetics Ft = K_0_t (7)
where Ft = fraction of drug released at time t, and K_0_ = zero order release constant
First order kinetics ln (1 − F) = K_1_t (8)
where F = fraction of drug released at time t, and K_1_ = first order release constant
Higuchi Model F = K_2_t^1/2^
(9)
where F = fraction of drug released at time t, and K_2_ = Higuchi constant
Korsmeyer – Peppas model Mt/M_∞_ = K_3_t^n^
(10)
where Mt/M_∞_ = fraction of drug release at time t, n = release exponent, and K_3_ = rate constant.

### 4.14. Statistical Analysis

The statistical analysis was performed for all the experiments by using Student’s *t*-test in order to determine the differences between the tests, which were considered statistically significant because the obtained *p*-value was less than 0.05.

## Figures and Tables

**Figure 1 gels-08-00155-f001:**
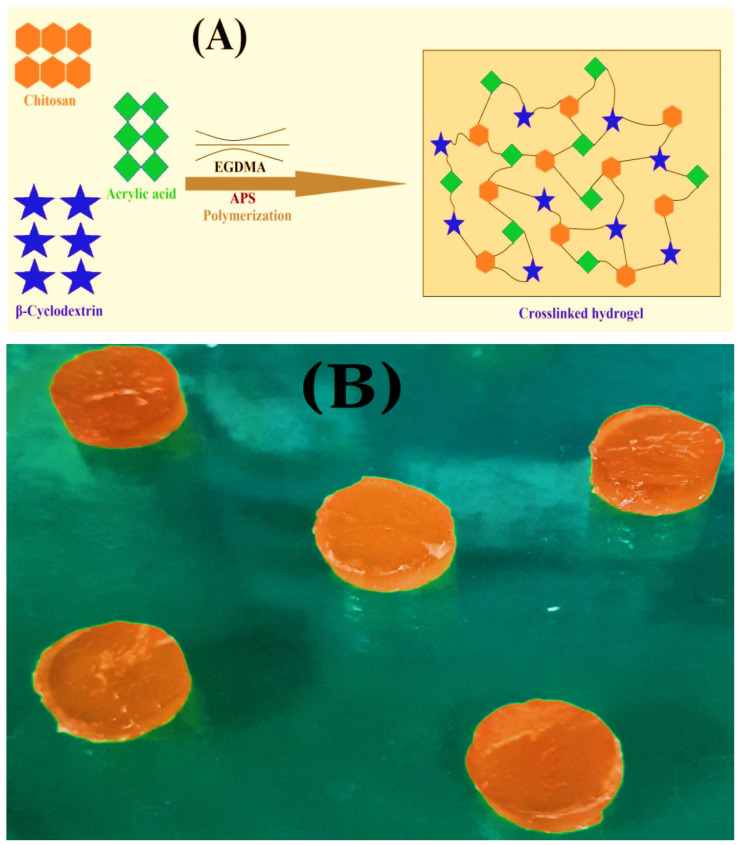
(**A**) Proposed schematic diagram, and (**B**) physical appearance of prepared CS/β-CDcPAa hydrogel.

**Figure 2 gels-08-00155-f002:**
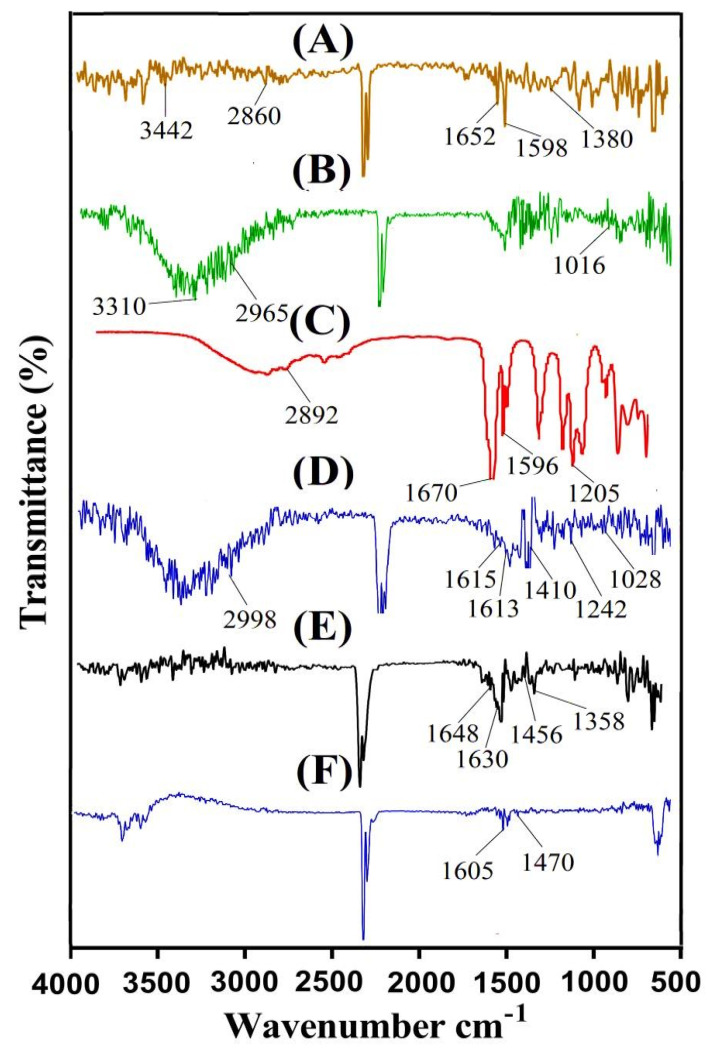
FTIR spectra of (**A**) CS, (**B**) β-CD, (**C**) Aa, (**D**) unloaded CS/β-CDcPAa hydrogel, (**E**) 5-ASA, and (**F**) loaded CS/β-CDcPAa hydrogel.

**Figure 3 gels-08-00155-f003:**
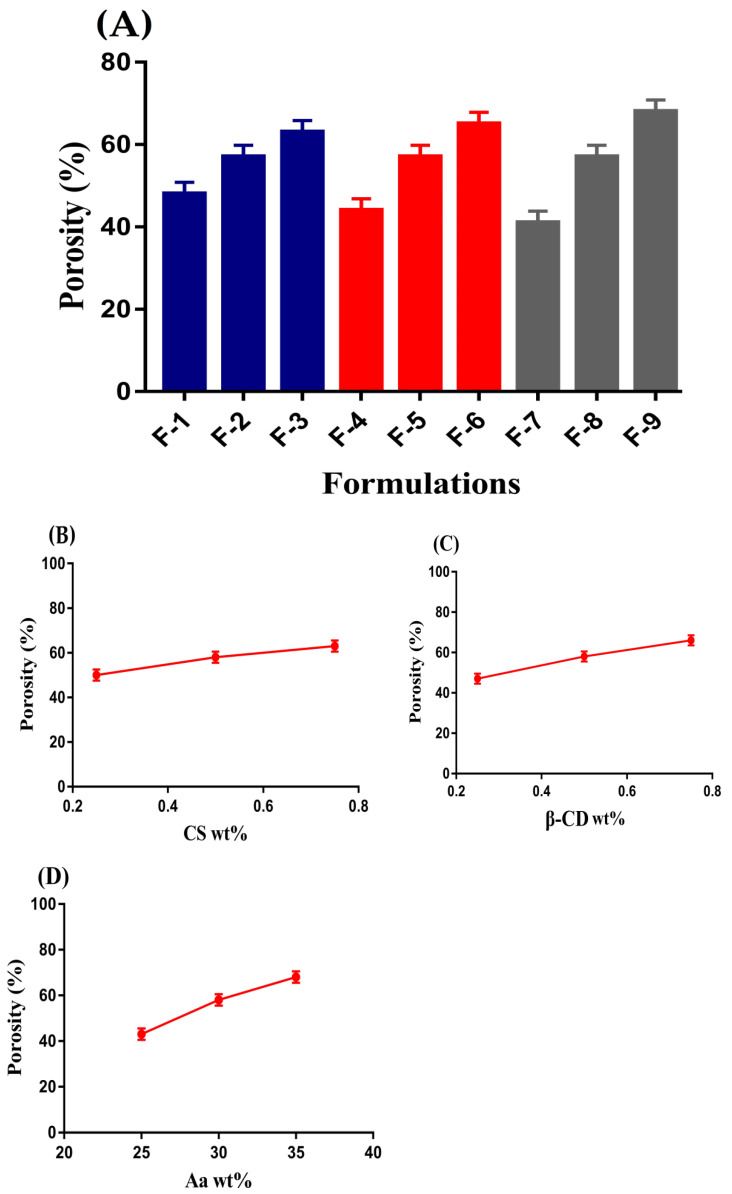
Effect of (**A**) all formulations, (**B**) CS, (**C**) β-CD, and (**D**) Aa on porosity of CS/β-CDcPAa hydrogel.

**Figure 4 gels-08-00155-f004:**
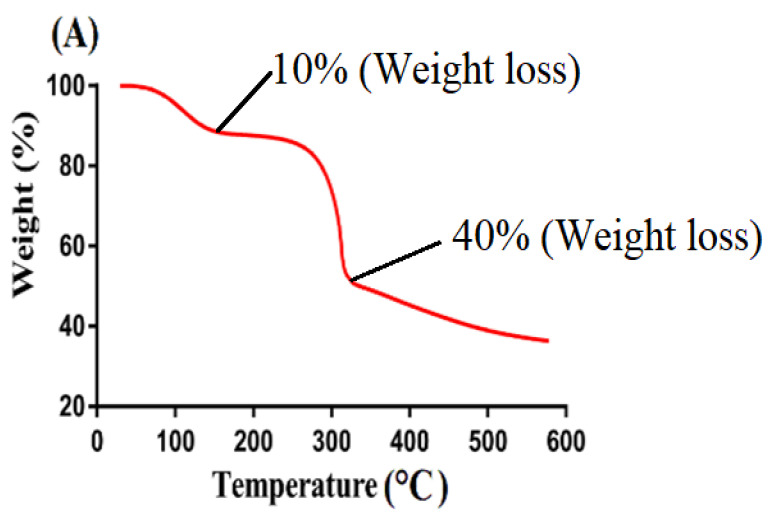
TGA of (**A**) CS, (**B**) β-CD, and (**C**) CS/β-CDcPAa hydrogel.

**Figure 5 gels-08-00155-f005:**
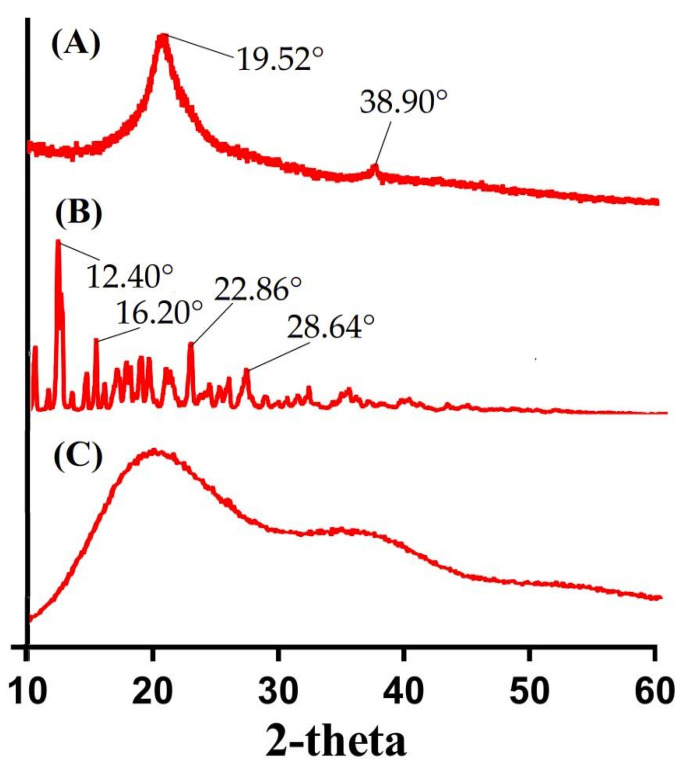
PXRD of (**A**) CS, (**B**) β-CD, and (**C**) CS/β-CDcPAa hydrogel.

**Figure 6 gels-08-00155-f006:**
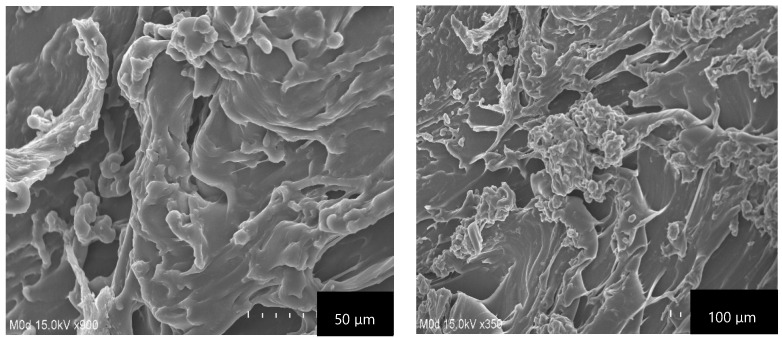
Scanning electron microscopy of CS/β-CDcPAa hydrogel.

**Figure 7 gels-08-00155-f007:**
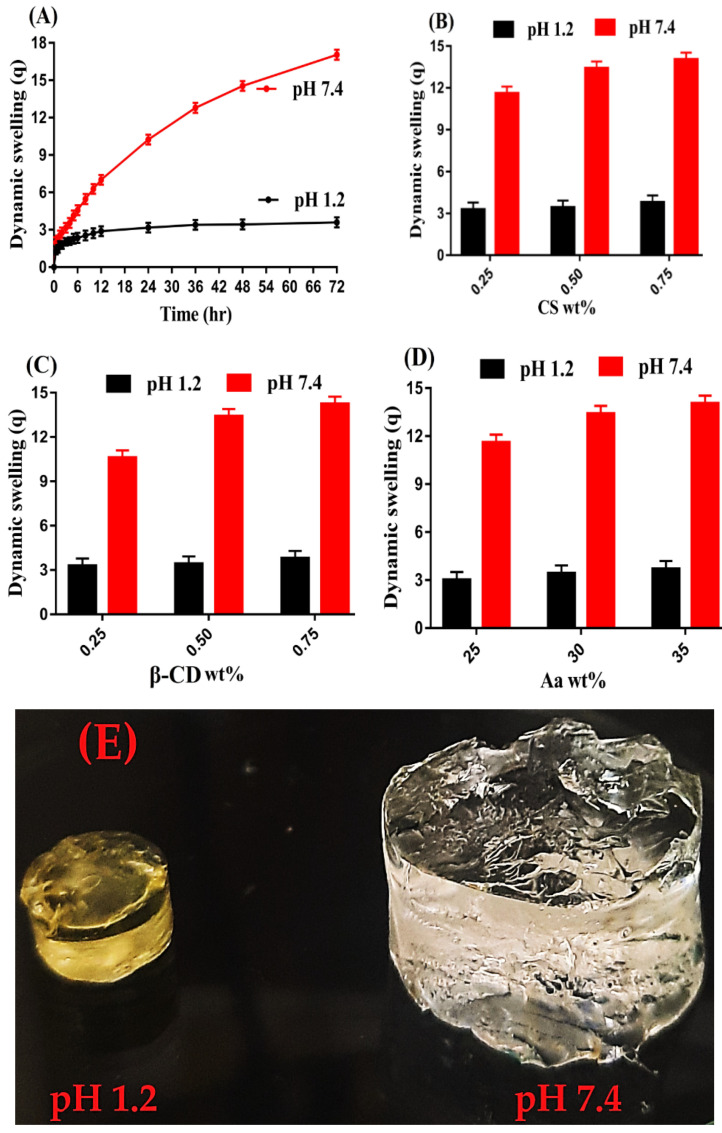
Effect of (**A**,) pH, (**B**) CS, (**C**) β-CD, and (**D**) Aa on dynamic swelling of CS/β-CDcPAa hydrogel, (**E**) physical appearance of swelled CS/β-CDcPAa hydrogel in pH 1.2 and 7.4.

**Figure 8 gels-08-00155-f008:**
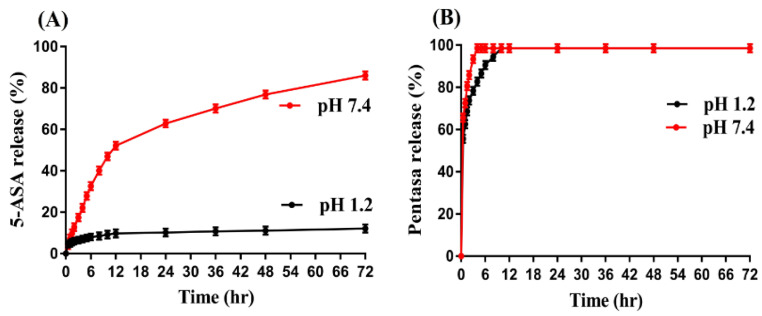
Effect of pH on (**A**) percent drug release from CS/β-CDcPAa hydrogel, (**B**) percent drug release from Pentasa, and effect of (**C**) CS, (**D**) β-CD, and (**E**) Aa on percent drug release from CS/β-CDcPAa hydrogel.

**Table 1 gels-08-00155-t001:** Feed ratio scheme for the formulation of CS/β-CDcPAa hydrogels.

F. Code	Polymer (CS) g/100 g	Polymer (β-CD) g/100 g	Monomer (Aa) g/100 g	Polymer Volume Fraction	
				pH 1.2	pH 7.4
F-1	0.25	0.50	30	0.300	0.083
F-2	0.50	0.50	30	0.282	0.077
F-3	0.75	0.50	30	0.271	0.073
F-4	0.50	0.25	30	0.304	0.086
F-5	0.50	0.50	30	0.282	0.077
F-6	0.50	0.75	30	0.267	0.070
F-7	0.50	0.50	25	0.308	0.093
F-8	0.50	0.50	30	0.282	0.077
F-9	0.50	0.50	35	0.263	0.066

APS and EGDMA were used 0.5 g/100 g throughout all formulations.

**Table 2 gels-08-00155-t002:** Sol–gel analysis and drug loading of CS/β-CDcPAa hydrogels.

Formulation Code	Sol	Gel	Drug Loaded (mg)/400 mg of Dry Gel	
	Fraction	Fraction		
	%	%	Weight Method	Extraction Method
F-1	10.60	89.40	125.32 ± 0.94	123.90 ± 0.98
F-2	09.21	90.79	132.63 ± 1.04	131.48 ± 0.94
F-3	08.72	91.28	137.28 ± 0.98	136.12 ± 1.04
F-4	11.52	88.48	121.84 ± 1.08	119.73 ± 0.91
F-5	09.21	90.79	132.63 ± 1.04	131.48 ± 0.94
F-6	07.90	92.10	140.87 ± 0.97	138.11 ± 1.05
F-7	11.80	88.20	114.54 ± 1.07	113.25 ± 0.97
F-8	09.21	90.79	132.63 ± 1.04	131.48 ± 0.94
F-9	08.94	91.06	144.82 ± 1.04	143.51 ± 1.14

**Table 3 gels-08-00155-t003:** Kinetic modeling release of 5-ASA from CS/β-CDcPAa hydrogels.

F. Code	Zero Order	First Order	Higuchi	Korsmeyer-Peppas	
	r^2^	r^2^	r^2^	r^2^	n
F-1	0.9364	0.9912	0.9876	0.9848	0.5470
F-2	0.9636	0.9974	0.9890	0.9753	0.5228
F-3	0.9598	0.9864	0.9740	0.9823	0.6054
F-4	0.9312	0.9853	0.9065	0.9739	0.5993
F-5	0.9636	0.9974	0.9890	0.9753	0.5228
F-6	0.9512	0.9932	0.9910	0.9810	0.6183
F-7	0.9765	0.9880	0.9612	0.9636	0.6310
F-8	0.9636	0.9974	0.9890	0.9753	0.5228
F-9	0.9390	0.9781	0.9543	0.9584	0.5864

## Data Availability

Not applicable.

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
