# Peer review of "Designing of pH-Sensitive Hydrogels for Colon Targeted Drug Delivery; Characterization and In Vitro Evaluation"

_gels, 2022, doi:10.3390/gels8030155_

Round 1
Reviewer 1 Report
This manuscript can be accepted for publication after the authors provide sufficient responses to the following comments:
- Since the hydrogel will be employed as the carrier for the drug delivery, why the TGA study was conducted?
- The SEM image of hydrogel provides no significant information about the structure of the hydrogel
- The kinetic equations should be written in the manuscript for easy reference.
Reviewer 2 Report
The manuscript “Designing of pH-sensitive hydrogels for colon targeted drug delivery; characterization and in-vitro evaluation” by Muhammad Suhail et.al., summarizes a detailed research work, to design and evaluate pH-sensitive hydrogels by free radical polymerization technique for targeted delivery of 5-aminosalicylic acid to colon. The concept and idea of the work is good, but this is a redundant work, I do not see any novelty in this work. Serval studies of this kind already exists. I would request the authors to revisit their manuscript and readdress all their data collected as tables and make visually appealing and easy to understand graphs and figures. A lot has been presented, but the main idea of it is missing. Readers do not have time to spend hours in a manuscript. And in such cases, the manuscript loses its scientific point, despite of the work being good like yours. I will proceed with a decision once changes are addressed.
However, Kindly revise as below:
- Kindly add a graphical/pictorial/structural representation (1) showing the cross linking of chitosan, β-Cyclodextrin, acrylic acid, ethylene glycol dimethacrylate, and Ammonium persulfate. This manuscript is built on this concept and this important piece of explanation is missing. Also, explain how these hydrogels change in terms of shape (swelling) at the different pH, as release profile of the drug is based on this pH kinetics.
- Section 1. Introduction, kindly add why this pH ranges were chosen, although it’s evident it’s for colon, but this should be mentioned in the introductory section.
- Figure 1, Line 142. Kindly add the bonds (C-C, C=N etc.) or short description on the figure/spectra itself, figures should be self-explanatory, no readers have time to scroll up and down of the manuscript. The way figures are presented adds a lot of attention to readers.
- Table 2. Where is the picture of the hydrogel that was made? Kindly add figures along with the tables to help explain better, visual representation is needed to justify results
- Figure 2, Line 179. Kindly add figures, along with the graphs. Also add statistical analysis and p* value for each. In any biological study, statistical analysis is a must.
- Figure 3, Line 202. What does the graph imply, what does the shoulder peak imply? Kinldy make figures self-explanatory. Also, why this range of 100-600 degree Celsius was chosen?, is this needed, our body temperature is 37 degree Celsius, why this range is not chosen for the study ?
- Line 185, “……120°C due to the evaporation of water…….”At this temperature, water will evaporate, its’s a known fact, what is gained out of this study, by doing thermal study of a hydrogel at that temperature? Stability of a hydrogel or any pharmaceutical formulation is evaluated at an elevated temperature of 40 degree Celsius with relative humidity of 75.
- Line 204, XRD analysis is done to understand the form change in comparison to its initial form. Kindly add this line and reference it. Again, Figure 4, is just a graph/peak with no information, kindly add the peak values, explain what is changing at that 2-theta value, what bonds do you anticipate to change, was EDAX done, if yes, kindly add the elemental change in compositions to help support the results.
- Figure 5, SEM. The figure makes no sense to me. Add a figure or do SEM analysis, of an intact hydrogel vs swelled up hydrogel, and then compare the fibers before and after, this will be informative. Use the same hydrogel to do before/after study get this kind of result. The current figure is not informative.
- Figure 6. Effect of pH should be supported by visual hydrogel study.
- Section 2. Results. Kindly address these few points before discussion of each section or analysis or tests? Each of the below listed questions should be the way to address any result or discussion. This way the readers can understand the propose or motto of doing that study rather than presenting your results.
- What is the purpose or goal of this testing?
- What information does this study give about the hydrogel?
- What methods is chosen to evaluate, this?
- What references were used to support the results?
Kindly re-address each result/test section with the above questions.
- Kindly elaborate more on the introduction sections, and the result and discussion sections with references. Appropriate references are required to support the claim and results of any study.
Reviewer 3 Report
Respected Authors
Many thanks for your efforts done during this work.
There are some minor questions regarding this work.
1- We need the specifications of chitosan CS used during this work.
2- In section 4.2 preparation of CS/B-CDcPAa hydrogels, in line 364, stirred for 2 min - what is the rate of stirring (rpm).
3- In section 4.2 preparation of CS/B-CDcPAa hydrogels, in line 366, what are the concentrations of EGDMA and APS during work.
4- In characterization of the prepared hydrogels, through FT-IR, Thermal analysis and PXRD, why you did not perform the analysis for pure drug. It is supposed to be included within the analysis to study the changes underwent for the drug, therefore, it is recommended to repeat this studies with drug.
5- Why you did not perform DSC study to detect the effect of formulation on melting point of the components?
6-In fig 55,it is recommended to do the same study for loaded hydrogel.
Round 2
Reviewer 2 Report
Kindly accept